# Chemical Composition, Antioxidant and Cytotoxicity Activities of Leaves, Bark, Twigs and Oleo-Resin of *Dipterocarpus alatus*

**DOI:** 10.3390/molecules24173083

**Published:** 2019-08-25

**Authors:** Chawalit Yongram, Bunleu Sungthong, Ploenthip Puthongking, Natthida Weerapreeyakul

**Affiliations:** 1Biomedical Science Program, Graduate School, Khon Kaen University, Khon Kaen 40002, Thailand; 2Pharmaceutical Chemistry and Natural Products Research Unit, Faculty of Pharmacy, Mahasarakham University, Mahasarakham 44150, Thailand; 3Division of Pharmaceutical Chemistry, Faculty of Pharmaceutical Sciences, Khon Kaen University, Khon Kaen 40002, Thailand

**Keywords:** anticancer, antioxidant, polyphenol, sesquiterpenes, GC-MS analysis, *Dipterocarpus*

## Abstract

*Dipterocarpus alatus* (Dipterocarpaceae) is a medicinal plant whose use is well known for the treatment of genito-urinary diseases. However, there is no report of its cytotoxic potential. In this study, the chemical composition, antioxidant and cytotoxic activities of extracts of the leaves, bark, twigs and oleo-resin from *D. alatus* are investigated. Cytotoxicity was measured by the neutral red (NR) assay against HCT116, SKLU1, SK-MEL2, SiHa and U937 cancer cell lines and antioxidant capacity was evaluated by DPPH, ABTS radical scavenging, and ferric reducing antioxidant power (FRAP) assays. The chemical composition was analyzed by gas chromatography–mass spectrometry (GC-MS). Leaf, bark and twig extracts exhibited stronger antioxidant activity than oleo-resin, with bark extract showing the highest antioxidant activity and the highest total phenolic content. All samples showed more cytotoxic activity against the U937 cell line than HCT116, SKLU1, SK-MEL2 and SiHa cells with oleo-resin being more cytotoxic than melphalan against U937 cells. Chemical composition analysis of oleo-resin by GC-MS showed that the major components were sesquiterpenes, namely α-gurjunene (30.31%), (-)-isoledene (13.69%), alloaromadendrene (3.28%), β-caryophyllene (3.14%), γ-gurjunene (3.14%) and spathulenol (1.11%). The cytotoxic activity of oleo-resin can be attributed to the sesquiterpene content, whereas the cytotoxic and antioxidant activities of leaf, bark and twig extracts correlated to total phenolic content.

## 1. Introduction 

The most common incidences of cancer in Thailand are breast, cervix, colorectal, liver and lung cancers which made up 59.2% of the cancer burden in 2012, including 4% for the incidence of leukemia [1] and 4% incidence of melanoma which is mostly responsible for skin cancer [2]. Nowadays, cancer treatment is very costly, requiring expensive facilities, highly specialized health personnel and expensive drugs. Various approaches have been used for cancer treatment in addition to chemotherapy such as allopathic drugs for chemoprevention [3]. Moreover, chemotherapy generally leads to drug resistance and severe side effects [4]. Between 1940 and 2002, about 40% of all available anticancer drugs were natural products or natural products derived with 8% of those considered natural product mimics [5]. Thus, medicinal plants have made a tremendous contribution to the development of new drugs against diseases, including cancer [6].

Plants of genus *Dipterocarpus* have been reported to have many bioactivities including antibacterial, antioxidant [7], cytotoxic [8], anti-inflammatory [9] and anti-filarial activities [10]. Extracts of the bark and leaves of *D. turbinatus* used in Ayurvedic and Unani medicines showed potent cytotoxic activity against breast cancer cell lines [11]. The phytochemical constituent of plant genus *Dipterocarpus* produces a high yield of resveratrol oligomers (oligosilbenoids) [12], sesquiterpenes and triterpenes [13]. Ursolic acid, corosolic acid and isofouquierone triterpenes isolated from stems of *D. obtusifolius* showed cytotoxic activity against HepG2 and SK-OV-3 cancer cells at levels higher than chemotherapy Adriamycin [8]. Resveratrol oligomers isolated from the bark of *D. hasseltii*, inhibited murine leukemia P-388 cells, with hopeaphenol showing the strongest cytotoxic activity [14]. Vaticaffinol, a resveratrol oligomer isolated from *D. alatus* twigs and branches, reduced levels of serum uric acid and showed anti-inflammatory activity [15] while hopeahainol A and the oligostilbenoid, dipterocarpol A isolated from *D. alatus* stems exhibited acetylcholinesterase inhibitory activity [16]. 

The ethnomedicinal applications of *D. alatus* include the treatment of genitourinary diseases using oleo-resin [17] and *D. alatus* bark has been used in the treatment of liver diseases and rheumatism [18], to invigorate health, expel impurities and mitigate toothache [19] and to treat various skin diseases [20]. *D. alatus* balsam has been used for the treatment of gonorrhea and young *D. alatus* plants have been used to treat rheumatism and liver complaints [21]. However, there has been no report on the antioxidant and anticancer activity of *D. alatus* and the chemical composition has not been fully elucidated. The primary aim of the present work is to investigate the potential anticancer and antioxidant activities of various plant parts of *D. alatus* and subsequently the chemical characterization of the given parts using gas chromatography–mass spectrometry (GC-MS) in an effort to underline the correlation between chemical content and biological activity. 

## 2. Results and Discussion

### 2.1. Extraction and Phytochemical Analysis

Different parts of *D. alatus*, like leaves, twigs and bark were macerated with methanol to yield extracts weighing 153.48 g (11.80% yield), 115.41 g (6.07%) and 29.68 g (4.24%), respectively. The qualitative phytochemical screening was carried out to identify the presence of secondary metabolites including alkaloids, steroids, tannins, xanthones, and reducing sugars (Table 1). Tannins and reducing sugars were found in leaf, bark and twig extracts. Alkaloids and xanthones were found only in bark, while steroids were found in bark, twigs and oleo-resin.

### 2.2. Chemical Constituents of D. alatus Extracts

The major bioactive secondary metabolite of plants in the genus *Dipterocapus* is sesquiterpenoid essential oil containing sesquiterpenes as its major component [22]. Therefore, we characterized the secondary metabolites from the various parts of *D. alatus* by GC-MS. The oleo-resin elution profile identified 20 compounds (Table 2) and the six major compounds are summarized in Figure 1. Compounds **1**–**18** were identified to be members of the sesquiterpene family and compounds **19**–**20** were from the triterpene family. The six most abundant compounds were all sesquiterpenes with α-gurjunene as the major component, comprising 30.31% of the total, followed by (-)-isoledene (13.69%), alloaromadendrene (3.28%), β-caryophyllene (3.14%), γ-gurjunene (3.14%) and spathulenol (1.11%). While α-gurjunene has previously been reported to be the major sesquiterpene component of *D. hasseltii* (36.00%), *D. intricatus* (80.00%) and *D. kerrii* (79.17%), it is a minor component in *D. retusus* (2.00%) and absent from *D. grandifloras* [22,23]. Sesquiterpenes and triterpenes were not detected in the *D. alatus* leaf and bark extracts, but a small amount (0.05%) of the triterpene loupenone was found in twig extracts.

### 2.3. Cytotoxic Activity

The cytotoxicity of the leaf, bark and twig extracts and oleo-resin exhibited similar results in all cancer cell lines. All *D. alatus* samples showed high cytotoxic activity against the U937 cell line. IC_50_ values of 91.3 ± 6.2 (leaf), 106.1 ± 7.8 (bark), 128.9 ± 2.5 (twig), and 63.3 ± 2.1 µg/mL (oleo-resin) were lower than the IC_50_ value for standard anticancer melphalan (228.4 ± 8.8 µg/mL) in Table 3. Oleo-resin showed cytotoxic activity against the HCT116, SK-LU-1, SK-MEL-2 and SiHa cell lines, but the IC_50_ values for oleo-resin were always higher than those for melphalan. In contrast, leaf and twig extracts showed little cytotoxic activity against the HCT116, SK-LU-1, SK-MEL-2 and SiHa cell lines and bark extract showed only moderate cytotoxicity against the SK-LU-1 and SiHa cell lines. However, oleo-resin was toxic to normal Vero cells (IC_50_ 88.7 ± 4.1 µg/mL), while leaf, bark and twig extracts did not exert any toxic effect to normal Vero cells. The chemotherapy sensitivity and treatment outcomes of any tested compounds can be varied between differing tumor cell types. Taken together, these results indicate that the *D. alatus* leaf, bark and twig extracts have high selective cytotoxic activity against U937 cells with selective index (SI) values of 5.5, 4.7 and 3.9, respectively (Table 4). A selective index value higher than three in this case indicates high selectivity for cancer cells [24]. While the oleo-resin showed high toxicity to U937 cells, its cytotoxic selectivity (SI 1.4) is only moderate due to its toxicity to normal Vero cells. The cytotoxicity of oleo-resin against U937 might be related to its sesquiterpene chemical composition as (−)-isoledene has been shown to have cytotoxic activity against HCT 116 [25], alloaromadendrene has been shown to have antiproliferative activity against the highly malignant +SA mouse mammary epithelial cell line [26], β-caryophyllene exhibited cytotoxic activity against HCT 116 cells via apoptosis cell death pathway [27], and spathulenol showed antiproliferative activity against MCF-7, OVCAR-3 and HaCaT cells [28]. This finding suggests that compounds which belong to sesquiterpenes composition in oleo-resin of *D. alatus* are responsible for the anticancer properties.

### 2.4. Antioxidant Activities and Total Phenolic Contents

The radical scavenging activities of the *D. alatus* extracts and oleo-resin were evaluated using DPPH and ABTS radical scavenging assays and reducing power was determined using the ferric reducing antioxidant power (FRAP) assay. As shown in Table 5, the results from the DPPH and ABTS assays indicate no antioxidant activity for oleo-resin (IC_50_ > 1000 µg/mL, both) while the bark extract showed similar radical scavenging capacity to Trolox (IC_50_ 5.76 ± 0.19 and 9.37 ± 0.03 µg/mL compared to 3.93 ± 0.02 and 10.20 ± 0.10 µg/mL, respectively). Extracts of twigs and leaves showed moderate radical scavenging activity (Table 5). The ferric reducing antioxidant power (FRAP) results show the extracts to have reducing power activity in the following order: barks > leaves > twigs > oleo-resin, but all extracts had much lower reducing power activity than the standard antioxidant Trolox.

Total phenolic contents (TPC) of crude extracts of *D. alatus* ranged from 15.14 to 366.43 mg gallic acid equivalents (GAE) g/g DW. The bark extract showed the highest phenolic content (366.43 mg GAE/g DW) followed by leaves, twigs and oleo-resin, respectively (Table 5). These TPC results correlate with antioxidant activities, which might be related to the presence of tannin and reducing sugar groups in the extracts of barks, leaves and twigs.

## 3. Materials and Methods

### 3.1. Plant Materials

The leaves, twigs, bark and oleo-resin of *D. alatus* were collected from Khon Kaen province, Thailand during June 2014. The *D. alatus* was identified by Suppachai Tiyaworanant and a voucher specimen was deposited in the Division of Pharmacognosy and Toxicology, Faculty of Pharmaceutical Sciences, Khon Kaen University, under registration NO. PSKKF03682.

### 3.2. Sample Preparation

The leaves, twigs and bark of *D. alatus* were cut and mashed to dry powder (leaves 1300 g, twigs 700 g, and bark 1900 g). The dry powder of each part was macerated with methanol for 24 h at room temperature and repeated three times. The combined organic solvent was removed after filtration and evaporated under reduced pressure followed by freeze drying to give crude extract. The oleo-resin crude was taken from the *D. alatus* tree. The sample was heated with temperature from 40–60 °C before use.

### 3.3. Phytochemical Test

The crude extracts of *D. alatus* were screened for the presence of tannins, xanthones, steroids, alkaloids and reducing sugars according to methods modified from Sasidharan et al. and Manosroi et al. [29,30]. Briefly, 30 mg of crude extract was dissolved in 2 mL of ethanol and 2 mL of 15% FeCl_3_ was added to the supernatant. Development of a blue-black or dark green color indicated tannins were present. For xanthones, 30 mg of crude extract was dissolved in 2 mL of ethanol. Following centrifugation, 100 µL of 5% KOH was added to the supernatant and the presence of xanthones was observed by the development of a yellow precipitate. The presence of steroids was tested by adding 1 to 2 drops of conc. H_2_SO_4_ to 30 mg of crude extract dissolved in 2 mL of chloroform. Steroids produced a red color in the lower chloroform layer. For alkaloids, 30 mg of extract was dissolved in 1 mL of ethanol and mixed with 2 mL 1% HCl and 6 drops of Dragendorff’s reagent. A reddish-brown precipitate with turbidity indicated that alkaloids were present. For reducing sugars, 30 mg of crude extract was mixed with 1 mL Fehling’s solution. The mixture was heated in a water bath for 10 min. The presence of reducing sugars was observed as a brick-red precipitate.

### 3.4. Antioxidant Activity

#### 3.4.1. DPPH Radical Scavenging

The DPPH assay was modified from Ghasemzadeh et al. [31]. DPPH was dissolved in ethanol at a concentration of 200 µM. The 10 mg/mL stock solution of crude extract was prepared in MeOH. Crude extract solution at 10–500 µg/mL and 200 μM DPPH (100 μL each) were added into 96-well plates. The mixture was incubated at room temperature for 30 min in the dark. After incubation, absorbance was read at 490 nm using a microplate reader (Sunrise™, Grödig, Austria). The IC_50_ describes the concentration that causes half-maximal inhibition by the extract. Trolox was used as the positive control.

#### 3.4.2. ABTS Radical Scavenging

This assay was modified from Kim et al. [32]. The ABTS radical cation was prepared by mixing 28 mM ABTS with potassium persulfate (2.45 mM) in purified water. The ABTS reagent was kept at room temperature in the dark for 18 h before use. The ABTS (150 μL) was mixed with crude extract at 10–500 µg/mL (50 μL) in 96-well plates. Absorbance of the mixtures was read at 415 nm using a microplate reader and the IC_50_ was determined. Trolox was used as the positive control.

#### 3.4.3. Ferric Reducing Antioxidant Power (FRAP)

A mixture of 300 mM acetate buffer pH 3.6, 20 mM FeCl_3_ solution and 10 mM TPTZ in 40 mM HCl in the ratio of 10:1:1 was prepared for the FRAP reagent. Crude extracts at 500 µg/mL (10 μL) were mixed with ultrapure water (30 μL) and FRAP reagent (300 μL) in 96-well plates and incubated at 37 °C for 4 min. The absorbance was read at 595 nm using a microplate reader. Trolox was used as the positive control. The FRAP is reported in (mmole)/100 g DW units. The assay was modified from Benzie and Strain [33]. 

#### 3.4.4. Total Phenolic Content (TPC)

The total phenolic content of the *D. alatus* extracts were determined using the method of Kaisoon et al. [34]. Folin–Ciocalteu reagent was diluted 10-fold with distilled water before use. A 16 μL aliquot of 500 µg/mL extract was mixed with 117 μL of Folin–Ciocalteu reagent in 96-well plates at room temperature for 5 min. Then, sodium carbonate (60 g/L) solution was added. The absorbance was read at 700 nm using a microplate reader after incubation at room temperature for 90 min. The total phenolic content is shown as gallic acid equivalents (mg GAE/g DW). 

### 3.5. Cell Culture

The normal African green monkey kidney (Vero), human colon cancer (HCT116), melanoma (SK-MEL-2), lung adenocarcinoma (SK-LU-1) and cervix adenocarcinoma (SiHa) cell lines were cultured in Dulbecco’s Modified Eagle Medium (DMEM) supplemented with 10% fetal bovine serum (FBS). The human leukemic U937 cell line was cultured in RPMI 1640 medium with 10% FBS. The media were supplemented with 1% penicillin/streptomycin and cells were maintained at 37 °C in 5% atmospheric CO_2_ humidified incubator. 

### 3.6. Cytotoxic Activity

The cytotoxic activity of the crude extracts was assessed using the neutral red (NR) assay. The exposure time was 24 h. Briefly, cells were seeded in 96-well plates (3 × 10^5^ cells/mL for Vero and HCT116; 4 × 10^5^ cells/mL for SK-MEL-2, SK-LU-1 and SiHa, and 5 × 10^5^ cells/mL for U937) and test samples at 10–500 µg/mL were added. After incubation under normal conditions for 24 h, cells were washed with PBS and 50 μg/mL of NR solution was added and plates were incubated for a further 2 h. Cells were washed again with PBS and 0.33% HCl in isopropanol was added. The absorbance was measured using a microplate reader at 537 nm for NR and at 650 nm for a reference wavelength [35]. The IC_50_ was calculated from a plot of % cytotoxicity and sample concentration. 

### 3.7. Gas Chromatography–Mass Spectrometry (GC-MS) Analysis 

The oleo-resin of *D. alatus* was analyzed by GC-MS with a SHIMADZU QP-2010 instrument (Kyoto, Japan). The oleo-resin was injected into an Agilent J&W VF-5ms column (30 m × 0.25 mm, 0.25 μm, Santa clara, CA, USA). The temperature of the injector was set at 290 °C. The oven temperature was initially set at 60 °C (hold time 3 min), with gradual increases in temperature from 60 to 120 °C (3.0 °C/min, hold 1 min), from 120 to 280 °C (2 °C/min, hold 1 min), and from 280 to 300 °C (10 °C/min, hold 2 min). Column flow was 1 mL/min. Carrier gas was helium 5.5, ionization energy was 70 eV and the split ratio was set to 1:30 after 45 s. The peaks were analyzed based on GC retention time and mass spectral identity was by comparison to the library (NIST147.LIB and WILEY7.LIB) [36].

### 3.8. Statistical Analysis

The antioxidant and cytotoxicity assays were carried out in triplicate. The results are presented as mean ± standard deviation. Statistical analysis was performed using SPSS 19.0 software (SPSS Inc., Chicago, IL, USA) for Windows. The results were analyzed using one-way analysis of variance ANOVA with Tukey’s honest significant difference test for comparing mean significant differences between samples (*p* < 0.05). 

## 4. Conclusion

In this study, extracts of *D. alatus* leaves, barks and twigs showed stronger antioxidant capacity than oleo-resin and this was correlated with total phenolic content. Moreover, the leaf, bark and twig extracts inhibited the U937 cancer cell line without affecting normal Vero cells. Oleo-resin showed high non-selective cytotoxic activity against all tested cell lines with higher activity against the U937 cancer cell line than melphalan. The cytotoxic activity of oleo-resin appeared to correlate with the presence of sesquiterpenes, and especially α-gurjunene, which is found in abundance in its balsam. In contrast, the cytotoxic and antioxidant activities of leaf, bark and twig extracts appeared to be related to the presence of phenolic compounds. Thus, these *D. alatus* extracts are promising targets for drug development in cancer and as antioxidant agents.

## Figures and Tables

**Figure 1 molecules-24-03083-f001:**
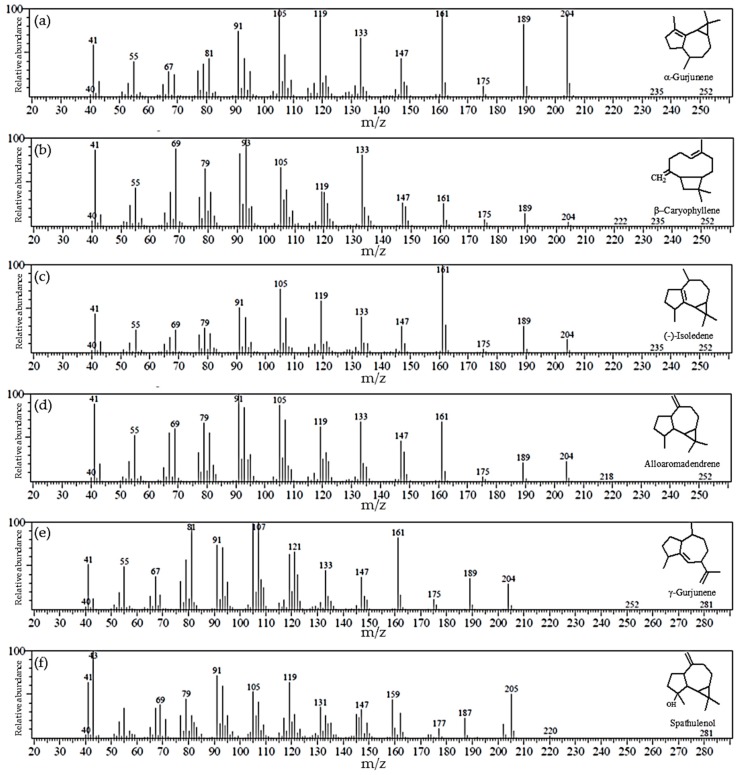
Gas chromatography-mass spectrometry (GC-MS) spectra of the major sesquiterpenes in oleo-resin. (**a**) α-Gurjunene; (**b**) (−)-isoledene; (**c**) alloaromadendrene; (**d**) β-caryophyllene; (**e**) γ-gurjunene; and (**f**) spathulenol.

**Table 1 molecules-24-03083-t001:** Phytochemical screening of *D. alatus* extracts.

Part of Plant	Chemical Group				
	Alkaloid	Steroid	Tannin	Xanthone	Reducing Sugar
Leaves	−	−	+	−	+
Bark	+	+	+	+	+
Twigs	−	+	+	−	+
Oleo-resin	−	+	−	−	−

Presence (+) and absence (−).

**Table 2 molecules-24-03083-t002:** Principal compound profiling of oleo-resin of *D. alatus* by GC-MS.

Compound	Retention Time	Compound Name	Molecular Weight	Molecular Formula	% (Peak area)	Fragmentation Pattern *m/z* (Decreasing Order of Abundance)
**1**	23.48	δ-Elemene	204	C_15_H_24_	0.08	121(100), 93, 136, 107, 79, 41, 161, 55, 67, 189, 204
**2**	24.05	α-Cubebene	204	C_15_H_24_	0.03	105(100), 119, 161, 91, 81, 41, 55, 204, 133
**3**	25.25	α-Copeane	204	C_15_H_24_	0.35	119(100), 105, 161, 93, 81, 41, 55, 133, 204, 65
**4**	25.98	β-Elemene	204	C_15_H_24_	0.78	93(100), 81, 68, 107, 41, 55, 121, 147, 133, 161, 189, 175, 204
**5**	26.87	α-Gurjunene	204	C_15_H_24_	30.31	105(100), 119, 161, 204, 189, 91, 133, 41, 147, 81, 55, 67
**6**	27.10	β-Caryophyllene	204	C_15_H_24_	3.14	93(100), 69, 41, 133, 105, 79, 55, 119, 147, 161, 189, 175, 204
**7**	27.63	(−)-Isoledene	204	C_15_H_24_	13.69	161(100), 105, 119, 91, 41, 133, 189, 147, 69, 69, 55, 204
**8**	28.26	α-Humulene	204	C_15_H_24_	0.94	93(100), 80, 121, 41, 147, 107, 67, 55, 204, 136
**9**	28.55	Alloaromadendrene	204	C_15_H_24_	3.28	91(100), 41, 105, 161, 133, 79, 119, 65, 55, 147, 204, 189
**10**	28.94	γ-Gurjunene	204	C_15_H_24_	3.14	107(100), 81, 161, 91, 121, 41, 55, 133, 67, 189, 147, 204, 175
**11**	29.65	Viridiflorene	204	C_15_H_24_	0.61	107(100), 93, 41, 119, 81, 133, 55, 161, 189, 67, 147, 204, 175
**12**	29.96	β-Vatirenene	202	C_15_H_22_	0.12	145(100), 105, 131, 91, 202, 120, 159, 187, 41, 77, 55, 67, 173
**13**	30.37	Selina-3,7(11)-diene	204	C_15_H_24_	0.52	122(100), 161, 107, 91, 81, 41, 55, 67, 133, 204, 189, 147
**14**	31.65	Calarene epoxide	220	C_15_H_24_O	0.04	41(100), 65, 93, 109, 119, 82, 135, 161, 185, 145, 177, 205, 220
**15**	31.85	Palustrol	222	C_15_H_26_O	0.13	122(100), 111, 81, 93, 41, 67, 161, 147, 189, 204, 133, 175, 222
**16**	32.17	Spathulenol	220	C_15_H_24_O	1.11	43(100), 91, 119, 41, 105, 205, 79, 159, 69, 131, 147, 187, 177, 220
**17**	32.30	(−)-Caryophyllene oxide	220	C_15_H_24_O	0.75	41(100), 79, 93, 69, 109, 121, 131, 161, 187, 177 202, 220
**18**	33.88	α-Cadinol	222	C_15_H_26_O	0.08	95(100), 43, 121, 161, 109, 204, 71, 58, 41, 179, 222
**19**	61.78	Otochilone	424	C_30_H_48_O	0.17	409(100), 65, 109, 257, 95, 55, 41, 311, 81, 149, 119, 271, 245, 297, 231, 173, 161, 204, 187, 424
**20**	62.68	Lupenone	424	C_30_H_48_O	0.38	95(100), 109, 205, 81, 55, 121, 69, 135, 149, 189, 41, 161, 175, 424, 128, 313, 245, 355

**Table 3 molecules-24-03083-t003:** Cytotoxic activity of *D. alatus* extracts against several cancer cells.

Samples	IC_50_ (µg/mL)					
	Vero	HCT116	SK-LU-1	SK-MEL-2	SiHa	U937
Leaves	>500 ^c^	>500 ^d^	>500 ^d^	>500 ^c^	>500 ^d^	91.3 ± 6.2 ^b^
Bark	>500 ^c^	>500 ^d^	273.0 ± 18.3 ^b^	>500 ^c^	197.8 ± 15.5 ^c^	106.1 ± 7.8 ^c^
Twigs	>500 ^c^	440.6 ± 28.7 ^c^	> 500 ^d^	>500 ^c^	>500 ^d^	128.9 ± 2.5 ^d^
Oleo-resin	88.7 ± 4.1 ^a^	340.3 ± 21.1 ^b^	336.6 ± 6.7 ^c^	215.2 ± 10.9 ^b^	59.6 ± 1.7 ^b^	63.3 ± 2.1 ^a^
Melphalan	215.6 ± 3.7 ^b^	179.6 ± 12.2 ^a^	56.9 ± 0.3 ^a^	20.2 ± 2.1 ^a^	27.1 ± 0.8 ^a^	228.4 ± 8.8 ^e^

^a, b, c, d, e^ letters indicate significant differences between rows in the same column at *p* < 0.05.

**Table 4 molecules-24-03083-t004:** Selective index of *D. alatus* leaf, bark and twig extracts, and oleo-resin.

Samples	Selective Index (SI) *				
	HCT116	SK-LU-1	SK-MEL-2	SiHa	U937
Leaves	1.0	1.0	1.0	1.0	>5.5
Bark	1.0	>1.8	1.0	>2.5	>4.7
Twigs	>1.1	1.0	1.0	1.0	>3.9
Oleo-resin	0.3	0.3	0.4	1.5	1.4
Melphalan	1.2	3.8	10.7	8.0	0.9

* SI as IC_50_ value of normal cell/IC_50_ value of cancer cell.

**Table 5 molecules-24-03083-t005:** Total phenolic content and radical scavenging activity of *D. alatus* leaf, bark and twig extracts, and oleo-resin.

Samples	Antioxidant Capacity			Total Phenolic Content (mg GAE/g DW)
	DPPH IC_50_ (µg/mL)	ABTS IC_50_ (µg/mL)	FRAP (mmole/100 g DW)	
Leaves	26.76 ± 0.25 ^d^	13.56 ± 0.09 ^c^	124.80 ± 0.19 ^c^	308.60 ± 4.32 ^b^
Bark	5.76 ± 0.19 ^b^	9.37 ± 0.03 ^a^	300.67 ± 2.93 ^b^	366.43 ± 11.52 ^a^
Twigs	16.53 ± 0.61 ^c^	15.46 ± 0.02 ^d^	102.79 ± 1.01 ^d^	128.29 ± 3.89 ^c^
Oleo-resin	>1000 ^e^	>1000 ^e^	21.02 ± 0.19 ^e^	15.14 ± 0.62 ^d^
Trolox	3.93 ± 0.02 ^a^	10.20 ± 0.10 ^b^	771.70 ± 11.37 ^a^	-

^a, b, c, d, e^ letters indicate significant differences between rows in the same column at *p* < 0.05. GAE—gallic acid equivalents, FRAP—ferric reducing antioxidant power, ABTS—2,2′-azino-bis(3-ethylbenzothiazoline-6-sulfonic acid).

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
