# Peer review of "Chemical Composition, Antioxidant and Cytotoxicity Activities of Leaves, Bark, Twigs and Oleo-Resin of Dipterocarpus alatus"

_molecules, 2019, doi:10.3390/molecules24173083_

Round 1

Reviewer 1 Report

This manuscript describes the investigation of the chemical composition, antioxidant and cytotoxic activities of extracts of various parts of medicinal plant Dipterocarpus alatus. This manuscript is concise and well written. Given that this is the first report on the antioxidant and cytotoxic activities of D. alatus, the obtained information might be useful. However, the information on the chemical composition is relatively limited. For example, only the values of total phenolic contents of D. alatus crude extracts are shown in this manuscript. It is not clear what kinds of phenolic compounds are contained in D. alatus crude extracts. Therefore, further analysis is required. In addition, there are a few concerns:

line 86-88: The authors describe that sesquiterpenes and triterpenes were not detected in the leaf and bark extracts and that a small amount of the triterpne was found in twig extracts. However, methanol might not be appropriate to extract non-polar sesquiterpenes or triterpenes.

line 106: It is described that D. alatus leaf, bark and twig extracts have high selective cytotoxic activity against U937 cells. The authors could discuss why these extracts have high selective cytotoxic activity against U937 cells.

line 142-150: The authors could provide more information about how oleo-resin sample was prepared.

Fig. 1: The resolution of Fig. 1 is low. The authors should provide clearer chromatograms with clearer chemical structures. In addition, GC-MS total ion chromatograms (TIC) of oleo-resin sample should be shown.

Reviewer 2 Report

Comments to the manuscript molecules-574656 "Chemical composition, Cytotoxic and Antioxidant Activities of Dipterocarpus alatus".

Authors present the report of a research aimed to determine the chemical composition, antioxidant and cytotoxic activities of extracts of the leaves, bark, twigs and oleo-resin from D. alatus. Cytotoxicity was measured by the neutral red (NR) assay against some cancer cell lines and antioxidant capacity was evaluated by DPPH, ABTS, and FRAP assays. The chemical composition was analyzed by gas chromatography-mass spectrometry (GC-MS). Leaf, bark and twig extracts exhibited stronger antioxidant activity than oleo-resin, with bark extract showing the highest antioxidant activity and the highest total phenolic content. All samples showed more cytotoxic activity against the U937 cell line. Chemical composition analysis of the oleo-resin by GC-MS showed the major components were sesquiterpenes. The cytotoxic activity of oleo-resin correlated with the sesquiterpene content, whereas the cytotoxic and antioxidant activities of leaf, bark and twig extracts correlated to total phenolic content. The experiment was correctly designed and carried out. The manuscript is clearly written and may be published after some minor editorial changes.

The manuscript contains good scientific information on the possibilities of the species Dipterocarpus alatus to provide useful substances in the fields of the feed and food, medicinal extracts, and different useful substances.

Probably, the entire paper is a little too synthetic and authors may make a further effort in providing more details on the experiment description and in the materials and methods presented, as well as in the results presentation and discussion.

However, in my opinion, the introduction is complete and the bibliography appropriately cited.

Considering the manuscript in the perspective of my field of expertise (domestication of wild plants), the study provides good preliminary information useful to project a possible harvest of wild genetic resources to make a first field collection. The information on chemical composition permit to design a standard pool of chemical tests in order to have a preliminary evaluation of the candidate selections.

Other experts and chemistry specialists, probably, will be more severe in evaluating the laboratory methodologies. Nevertheless, in my opinion, these experimental details are of sufficient quality according to the research objectives.

Reviewer 3 Report

I reviewed the manuscript by Yongram et al., in which authors describe a. the chemical composition of different plant parts of Dipterocarpus alatus using GC-MS method and b. the potential antioxidant and anticancer activities of plant extracts using various methods and techniques (DPPH, FRAP, cytotoxic assays, etc). The experiments were carefully designed and conducted and the results are rationally explained. However, the presentation of their results seems problematic in various parts. Moreover, English are unfortunately very poor in several places in the article. Notably, there are several common syntax and grammatical errors in the text and some sentences are badly constructed.

In general, I would like to recommend the publication of this manuscript in Molecules after the following major issues are addressed:

In the Title:

Authors should mention the parts of the plant that they were used in their research.

In the Abstract section:

lines 14-15: “…is a medicinal plant for treatment of genito-urinary diseases.”

This sentence should be rephrased for clarity, i.e. “…is a medicinal plant that its use is well known for treatment of genito-urinary diseases.” 

lines 16-17: “This study investigated the chemical composition, antioxidant and cytotoxic activities of extracts of the leaves, bark, twigs and oleo-resin from D. alatus.”

Change to “In this study, the chemical composition, antioxidant and cytotoxic activities of extracts of the leaves, bark, twigs and oleo-resin from D. alatus are examined or investigated”

lines 27-28: “The cytotoxic activity of oleo-resin correlated with the sesquiterpene content,...”.

Should change to: “The cytotoxic activity of oleo-resin can be attributed to the sesquiterpene content...”

In the Introduction section:

lines 33-34: “In Thailand have accounts for the second largest proportion, which is the cervix, colorectal, and lung cancers are contribute to cancer...”

Authors should rephrase for clarity.   

lines 41-42: “The medical plants have made….”

Authors should add citations to support their point of view, i.e.

-Kyriaki   Hatziagapiou; Eleni Kakouri; George I. Lambrou; Eleni Koniari; Charalabos   Kanakis; Olti Alexandra Nikola; Margarita Theodorakidou; Konstantinos Bethanis and   Petros  A.  Tarantilis. Crocins, the Active Constituents of Crocus Sativus L. Stigmas, Exert Significant Cytotoxicity on Tumor Cells In Vitro. Curr. Cancer Ther. Rev. 2018, 14, 1–10.

-Seca, A.M.L.; Pinto, D.C.G.A. Plant Secondary Metabolites as Anticancer Agents: Successes in Clinical Trials and Therapeutic Application. Int. J. Mol. Sci. 2018, 19, 263.

lines 43: “Plants in genus…” to “Plants of genus…”

lines 62-64: “The idea was to identify… Chromatography–Mass Spectrometry (GC–MS).”

Should change to: “The primary aim of the present work is to investigate the potential anticancer and antioxidant activities of various plant parts of D. alatus and subsequently the chemical characterization of the given parts using Gas Chromatography–Mass Spectrometry (GC–MS) in an effort to underline the correlation between chemical content and biological activity.”

In the Results and discussion section:

line 67: “The various parts of D. alatus, which are leaves, twigs and bark …”

Change to: “Different parts of D. alatus, like leaves, twigs and bark …” 

line 77: “as the major component.” change “as its major component.”

lines 90-91: Authors should give titles to both chart axes (100% abundance, time or m/z) in Figure 1. The resolution of the image should be improved. 

lines 96-97: “The cytotoxicity of the leaf, bark, and twig extracts and oleo-resin demonstrated the same manner in all cancer cell lines.”

Change to: “The cytotoxicity of the leaf, bark, and twig extracts and oleo-resin exhibited similar results in all cancer cell lines.”

In the Material and methods section:

line 219: “gradients” to “gradual increases in temperature”

In the Conclusion section:

lines 235-236: “The cytotoxic activity of oleo-resin appeared to correlate with the presence of sesquiterpenes, with α-gurjunene the most abundant species”   

Change to “The cytotoxic activity of oleo-resin appeared to correlate with the presence of sesquiterpenes, and especially α-gurjunene, which is found in abundance in its balsam.” 

Other issues:

Instead of Tables 3 and 5, authors could present column charts in their place in order to underline some significant differences among their results.

Round 2

Reviewer 1 Report

The manuscript was correctly revised according to my comments.

Reviewer 3 Report

I consider the revised manuscript by Yongram et al. adequately improved in comparison to its original version. Authors made a lot of effort to clarify some issues about their experiments and improve the presentation of their experimental results. Thus, I would like to recommend the publication of this manuscript in Molecules in its present form.